# Global Structure Knowledge-Guided Relation Extraction Method for Visually-Rich Document

**Xiangnan Chen**[1][*], **Qian Xiao**[2], **Juncheng Li**[1][†], **Duo Dong**[1], **Jun Lin**[2],
**Xiaozhong Liu**[3], **Siliang Tang**[1]

[1] Zhejiang University [2] Alibaba Group [3] Worcester Polytechnic Institute
{xnchen2020,junchengli,22121222,siliang}@zju.edu.cn
{xiaoqian.xq,linjun.lj}@alibaba-inc.com, xliu14@wpi.edu

## Abstract

Visual Relation Extraction (VRE) is a powerful means of discovering relationships between entities within visually-rich documents. Existing methods often focus on manipulating entity features to find pairwise relations, yet neglect the more fundamental structural information that links disparate entity pairs together. The absence of global structure information may make the model struggle to learn long-range relations and easily predict conflicted results. To alleviate such limitations, we propose a **Gl**O**b**al **S**tructure knowledge-guided relation **E**xtraction (**GOSE**) framework. GOSE initiates by generating preliminary relation predictions on entity pairs extracted from a scanned image of the document. Subsequently, global structural knowledge is captured from the preceding iterative predictions, which are then incorporated into the representations of the entities. This "generate-capture-incorporate" cycle is repeated multiple times, allowing entity representations and global structure knowledge to be mutually reinforced. Extensive experiments validate that GOSE not only outperforms existing methods in the standard fine-tuning setting but also reveals superior cross-lingual learning capabilities; indeed, even yields stronger data-efficient performance in the low-resource setting. The code for GOSE will be available at https://github.com/chenxn2020/GOSE.

## 1 Introduction

Visually-rich document understanding (VrDU) aims to automatically analyze and extract key information from scanned/digital-born documents, such as forms and financial receipts (Jaume et al., 2019; Cui et al., 2021). Since visually-rich documents (VrDs) usually contain diverse structured information, Visual Relation Extraction (VRE), as

---

[*] Work done during an internship at DAMO Research, Alibaba Group.

[†] Corresponding author.

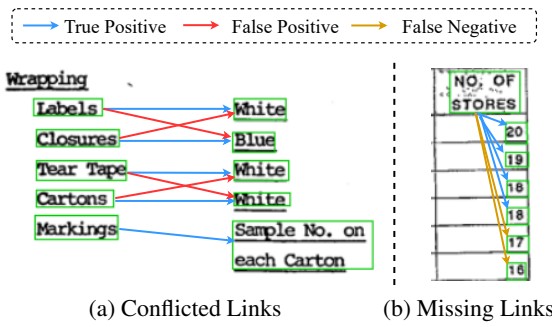

Figure 1: Incorrect relation predictions by the LiLT model on FUNSD.

a critical part of VrDU, has recently attracted extensive attention from both the academic and industrial communities (Jaume et al., 2019; Xu et al., 2021; Hwang et al., 2021; Wang et al., 2022a). The VRE task aims to identify relations between semantic entities in VrDs, severing as the essential basis of mapping VrDs to structured information which is closer to the human comprehension process of the VrDs (Zhang et al., 2021b). Recently, inspired by the success of pre-training in visually-rich document understanding (Li et al., 2021a; Xu et al., 2020; Wang et al., 2022a), many fine-tuning works predict relation for each entity pair independently, according to local semantic entity representations derived from the pre-trained model.

Although existing VRE methods have achieved promising improvement, they ignore global structure information, i.e. dependencies between entity pairs. **Without considering global structure knowledge, the model may easily predict conflicted links and struggle to capture long-range relations.** Taking the state-of-the-art model LiLT (Wang et al., 2022a) as an example, as shown in Figure 1(a), although each relational entity pair predicted by the LiLT may make sense semantically, there are conflicted relational entity pairs such as (*Labels*, *Blue*) and (*Closures*, *White*), as well as (*Tear Tape*, *White*) and (*Cartons*, *White*), whose link crosses each other from a global view.

This phenomenon indicates that methods only using local features do not have the sufficient discriminatory ability for conflicted predictions. Furthermore, as shown in Figure 1(b), even though LiLT accurately identifies the relational entity pairs (*No.OF STORES*, *19*) and (*No.OF STORES*, *18*), LiLT still hardly learns long-range relations such as (*No.OF STORES*, *16*) and (*No.OF STORES*, *17*). Our intuition is that global structure knowledge can help the model learn long-range relations. The model can predict the relational entity pair (*No.OF STORES*, *17*) by analogy with the global structure consistency between (*No.OF STORES*, *17*) and (*No.OF STORES*, *18*).

In this paper, we present the first study on leveraging global structure information for visual relation extraction. We focus on how to effectively mine and incorporate global structure knowledge into existing fine-tuning methods. It has the following two challenges: (1) **Huge Mining Space**. Considering N entities in a VrD, the computational complexity of capturing dependencies between entity pairs is quadratic to entity pair size ($N^2 \times N^2$). So it is difficult to mine useful global structure information in the lack of guidance. (2) **Noisy mining process**. Since the process of mining dependencies between entity pairs relies on initial entity representations and lacks direct supervision, global structure information learned by the model is likely to contain noise. Mined noisy global structure knowledge by a model can in turn impair the performance of the model, especially when the model has low prediction accuracy in the early training stage.

To this end, we propose a general global structure knowledge-guided visual relation extraction method named GOSE, which can efficiently and accurately capture dependencies between entity pairs. GOSE is plug-and-play, which can be flexibly equipped to existing pre-trained VrDU models. Specifically, we first propose a global structure knowledge mining (GSKM) module, which can mine global structure knowledge effectively and efficiently. The GSKM module introduces a novel spatial prefix-guided self-attention mechanism, which takes the spatial layout of entity pairs as the attention prefix to progressively guide mining global structure knowledge in a local-global way. Our intuition is the spatial layout of entity pairs in VrDs may be a valuable clue to uncovering global structure knowledge. As shown in Figure 1(a), we can recognize crossover between entity pairs in 2D space by computing the spatial layout of linking lines. Furthermore, in order to increase the robustness of GOSE and handle the noisy mining process, we introduce an iterative learning strategy to combine the process of entity representations learning and global structure mining. The integration of global structure knowledge can help refine entity embeddings, while better entity embeddings can help mine more accurate global structure knowledge.

In summary, the contributions of our work are as follows:

- We propose a global structure knowledge-guided visual relation extraction method, named GOSE. It can use the spatial layout as a clue to mine global structure knowledge effectively and leverage the iterative learning strategy to denoise global structure knowledge.

- GOSE can be easily applied to existing pre-trained VrDU models. Experimental results on the standard fine-tuning task over 8 datasets show that our method improves the average F1 performance of the previous SOTA models by a large margin: LiLT(+14.20%) and LayoutXLM(+12.88%).

- We further perform comprehensive experiments covering diverse settings of VRE tasks, such as cross-lingual learning, and low-resource setting. Experimental results illustrate advantages of our model, such as cross-lingual transfer and data-efficient learning.

## 2 Preliminaries

In this section, we first formalize the visually-rich document relation extraction task and then briefly introduce how the task was approached in the past.

### 2.1 Problem Formulation

The input to the VRE task is a scanned image of a document. Each visually rich document contains a set of semantic entities, and each entity is composed of a group of words and coordinates of the bounding box. We use a lower-case letter $e$ to represent semantic entity, where $e = \{[w^1, w^2, ..., w^k], [x^1, y^1, x^2, y^2]\}$. The sequence $[w^1, w^2, ..., w^k]$ means the word group, $x^1/x^2$ and $y^1/y^2$ are left/right x-coordinates and top/down y-coordinates respectively. The corresponding boldface lower-case letter **e** indicates its

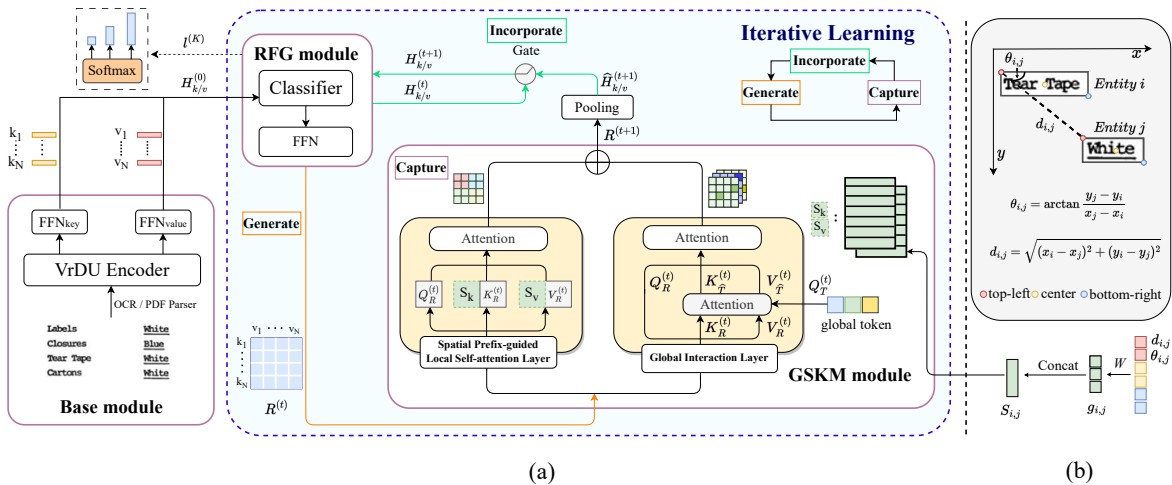

Figure 2: An illustration of the proposed GOSE framework. **(a)** The overview of GOSE. The framework consists of a Base module to generate initial *key/value* features based on entity representations from a pre-trained VrDU model, a relation feature generation (RFG) module to generate a relation feature for each entity pair, and a global structure knowledge mining (GSKM) module to mine global structure information in a local-global way. Besides, the RFG and GSKM modules are performed $K$ times in an iterative way to generate final predictions. **(b)** Spatial prefix construction. The spatial geometric features of entity pairs are computed as the spatial prefix to guide attention.

embedding. Let $\mathcal{E}$ and $\mathcal{R}$ represent the set of entities and relations respectively, where $\mathcal{E} = \{e_i\}_{i=1}^N$, $\mathcal{R} = \{(e_i, e_j)\} \subseteq \mathcal{E} \times \mathcal{E}$, $(e_i, e_j)$ mean the key-value entity pair and the directed link from $e_i$ to $e_j$. So each visually rich document can be denoted as $\mathcal{D} = \{\mathcal{E}, \mathcal{R}\}$. The goal of VRE task is to determine whether a relation (link) exists between any two semantic entities. Notably, the semantic entity may exist relations with multiple entities or does not have relations with any other entities.

## 2.2 Fine-tuning for Visual Relation Extraction

Inspired by the success of pre-training in visually-rich document understanding, most existing methods (Jaume et al., 2019; Xu et al., 2021; Wang et al., 2022a) fine-tune pre-trained VrDU model for VRE task. These methods take entity representations from a pre-trained VrDU model as input and then train a binary classifier to predict relations for all possible semantic entity pairs. Specifically, these methods project entity representations to *key/value* features respectively by two FFN layers. The features of *key* and *value* are concatenated and fed into a binary classifier to compute loss.

## 3 Methodology

In this section, we describe the proposed framework named GOSE in detail. As shown in Figure 2, our method consists of three main modules, a Base module, a Relation Feature Generation (RFG)

module, and a Global Structure Knowledge Mining (GSKM) module. In Section 3.1, we present how the Base module can be combined with a pre-trained VrDU model to generate initial *key/value* features. In Section 3.2, we introduce the RFG module to generate relation features. Then, in Section 3.3, we elaborate on how the GSKM module mines global structure knowledge in a local-global way. We further provide a theoretical analysis to better understand the efficient and effective features of the GSKM module. Finally, we show how to apply our iterative learning strategy to mine accurate global structure knowledge in Section 3.4.

### 3.1 Base Module

Given a visually rich document, the Base module first generates preliminary *key/value* features, which is same as most fine-tuning methods (Xu et al., 2021; Gu et al., 2022; Wang et al., 2022a). Specifically, we use a pre-trained VrDU model to obtain semantic entity representations. Then entity representation $\mathbf{e}$ is fed into two separated *Feed-Forward Networks* (*FFN*) to generate the initial *key* feature and *value* feature (denoted as $H_k^{(0)}$ and $H_v^{(0)}$ respectively), as written with Eq. (1):

$$
\begin{aligned}
H_k^{(0)} &= W_{key}\mathbf{e} + b_{key}, \\
H_v^{(0)} &= W_{value}\mathbf{e} + b_{value},
\end{aligned}
\tag{1}
$$

where $W_{key/value} \in \mathbb{R}^{2d_h \times d_h}$ are trainable weights and $b_{key/value} \in \mathbb{R}^{d_h}$ are trainable biases.

## 3.2 Relation Feature Gneration (RFG) Module

Taking *key/value* features as input, the RFG module generates relation features for all entity pairs. We denote *key/value* features at the $t$-$th$ iteration as $H_k^{(t)}$ and $H_v^{(t)}$ respectively, where $H_{k/v}^{(t)} \in \mathbb{R}^{N \times d_h}$. At the t-th iteration, the RFG module concatenates $H_k^{(t)}$ and $H_v^{(t)}$ as input and uses a bi-affine classifier to compute the relation *logits* of each entity pairs with Eq.(2):

$$l^{(t)} = H_k^{(t)} W_1 H_v^{(t)} + H_k^{(t)} W_2, \qquad (2)$$

where $l^{(t)} \in \mathbb{R}^{N \times N \times 2}$ denotes relation *logits* at the $t$-$th$ iteration. Then we employ a *FFN* block to generate a unified relation feature map (denoted as $R^{(t)} \in \mathbb{R}^{N \times N \times d_h}$), which contains relation features of all entity pairs. the relation feature map is calculated as follows:

$$R^{(t)} = W_r l^{(t)} + b_r, \qquad (3)$$

where $W_r \in \mathbb{R}^{2 \times d_h}$ is trainable weight and $b_r \in \mathbb{R}^{d_h}$ is trainable biases.

## 3.3 Global Structure Knowledge Mining (GSKM) module

The GSKM module mines the global structure knowledge on the relation feature map in a local-global way. As illustrated in Figure 2, this module consists of a spatial prefix-guided local self-attention (SPLS) layer and a global interaction layer. The SPLS layer first partitions the relation feature map into multiple non-overlapping windows and then performs spatial prefix-guided local self-attention independently in each window. The global interaction layer brings long-range dependencies to the local self-attention with negligible computation cost.

### 3.3.1 Spatial Prefix-guided Local Self-attention (SPLS) Layer

To address the challenge of mining global structure knowledge in a large computational space, we consider the spatial layout information of entity pairs in VrDs to guide self-attention computation. Most of existing VrDU methods independently encode the spatial information of each entity in the pre-training phase (i.e., the coordinate information of the bounding boxes in 2D space). However, it is necessary to insert spatial layout information of entity pairs explicitly in the fine-tuning phase (i.e.,

the spatial information of linking lines between bounding boxes in 2D space). Therefore, inspired by the success of prompt learning (He et al., 2022; Li et al., 2023a) we propose a spatial prefix-guided local self-attention (SPLS) layer to mine global structure knowledge in a parameter-efficient manner.

**Spatial Prefix Construction.** We calculate spatial geometric features (denoted as $S \in \mathbb{R}^{N \times N \times d_h}$) of linking lines between each entity pair as the spatial prefix. As shown in Figure 2(b), for an entity pair $(e_i, e_j)$, we calculate the direction and Euclidean distance of the line linking from the vertice $(x_i, y_i)$ of the bounding box of $e_i$ to the same vertice $(x_j, y_j)$ of $e_j$ as follows:

$$
\begin{aligned}
g_{i,j} &= [W_\theta \theta(i,j); W_d d(i,j)], \\
d(i,j) &= \sqrt{(x_i - x_j)^2 + (y_i - y_j)^2}, \\
\theta(i,j) &= \arctan \frac{y_j - y_i}{x_j - x_i},
\end{aligned}
\qquad (4)
$$

where $W_{\theta/d} \in \mathbb{R}^{1 \times \frac{d_h}{6}}$ are trainable weights. Therefore the spatial geometric features $S_{i,j}$ of entity pair $(e_i, e_j)$ is calculated as follows:

$$S_{i,j} = [g_{i,j}^{tl}; g_{i,j}^{ct}; g_{i,j}^{br}], \qquad (5)$$

where $g^{tl}$, $g^{ct}$, $g^{br}$ indicate top-left, center, and bottom-right points respectively. Then we treat $S$ as the spatial prefix and compute attention on the hybrid keys and values.

**Spatial Prefix-guided Attention.** We first partition $R^{(t)}$ into non-overlapping windows and then perform spatial prefix-guided local attention within each local window. The variant formula of self-attention with the spatial prefix as follows [1]:

$$
\begin{aligned}
\boldsymbol{R}_{local}^{(t)} &= \text{softmax}\big(\boldsymbol{R}^{(t)} \boldsymbol{W}_q [SW_k^s; \boldsymbol{R}^{(t)} \boldsymbol{W}_k]^\top\big) \begin{bmatrix} SW_v^s \\ \boldsymbol{R}^{(t)} \boldsymbol{W}_v \end{bmatrix} \\
&= \text{softmax}(\boldsymbol{Q}_R^{(t)} [S_k; \boldsymbol{K}_R^{(t)}]^\top) \begin{bmatrix} S_v \\ \boldsymbol{V}_R^{(t)} \end{bmatrix} \\
&= (1 - \lambda(\mathbf{R}^{(t)})) \underbrace{\text{Attn}(\boldsymbol{Q}_R^{(t)}, \boldsymbol{K}_R^{(t)}, \boldsymbol{V}_R^{(t)})}_{\text{standard attention}} \\
&\quad + \lambda(\mathbf{R}^{(t)}) \underbrace{\text{Attn}((\boldsymbol{Q}_R^{(t)}, S_k, S_v)}_{\text{spatial-prefix guided attention}},
\end{aligned}
$$

$$\qquad (6)$$

$$\lambda(\boldsymbol{R}^{(t)}) = \frac{\sum_i \exp(\boldsymbol{Q}_R^{(t)} S_k^\top)_i}{\sum_i \exp(\boldsymbol{Q}_R^{(t)} S_k^\top)_i + \sum_j \exp(\boldsymbol{Q}_R^{(t)} \boldsymbol{K}_R^{(t)\top})_j}, \qquad (7)$$

---

[1] Without loss of generality, we ignore the scaling factor $\sqrt{d}$ of the softmax operation for the convenience of explanation.

where $R_{local}^{(t)}$ refers to the local attention output at the t-th iteration, $\lambda(\boldsymbol{R}^{(t)})$ denotes the scalar for the sum of normalized attention weights on the key and value vectors from spatial prefix.

### 3.3.2 Global Interaction Layer (GIL)

After the SPLS layer effectively aggregates local correlations with window priors, we introduce a global interaction layer to bring long-range dependencies to the local self-attention. As illustrated in Figure 2, we use learnable global tokens $T \in \mathbb{R}^{M \times d_h}$ to compute the global interaction at the t-th iteration as follows:

$$
\begin{aligned}
\widehat{T}^{(t)} &= \text{Attn}(Q_T^{(t)}, \boldsymbol{K}_R^{(t)}, \boldsymbol{V}_R^{(t)}), \\
\boldsymbol{R}_{global}^{(t)} &= \text{Attn}(\boldsymbol{Q}_R^{(t)}, K_{\widehat{T}}^{(t)}, V_{\widehat{T}}^{(t)}),
\end{aligned}
\tag{8}
$$

where $R_{global}^{(t)}$ refers to the global interaction at the t-th iteration. $T$ will be updated in the same way as $H_{k/v}$ throughout the iterative learning process. Subsequently, we compute $R^{(t+1)}$ and employ the mean pooling operation to obtain the context-aware key and value features as:

$$
\begin{aligned}
\boldsymbol{R}^{(t+1)} &= \boldsymbol{R}_{local}^{(t)} + \boldsymbol{R}_{global}^{(t)}, \\
\widehat{H}_{k/v}^{(t+1)} &= \text{mean-pooling}(\boldsymbol{R}^{(t+1)}),
\end{aligned}
\tag{9}
$$

where $\widehat{H}_{k/v}^{(t+1)}$ contain global structure information.
**Analysis of GSKM.** Here we give some analysis to help better understand GSKM, especially effective and efficient features.

**Effectiveness.** GSKM can effectively learn global structure knowledge guided by spatial layout information in VrDs. As shown in Eq. 6, the first term $\text{Attn}(\boldsymbol{Q}_R^{(t)}, \boldsymbol{K}_R^{(t)}, \boldsymbol{V}_R^{(t)})$ is the standard attention in the content side, whereas the second term represents the 2D spatial layout guidelines. In this sense, our method implements 2D spatial layout to guide the attention computation in a way similar to linear interpolation. Specifically, the GSKM module down-weights the original content attention probabilities by a scalar factor (i.e., $1 - \lambda$) and redistributes the remaining attention probability $\lambda$ to attend to spatial-prefix guided attention, which likes the linear interpolation.

**Efficiency.** GSKM can reduce the computation complexity $N^4$ to $N^2 \times S^2$, where $S$ denotes window size. In the SPLS layer, with a relation feature map $R^{(t)} \in \mathbb{R}^{N \times N \times d_h}$ as input, we partition $R^{(t)}$ into non-overlapping windows with shape $(\frac{N}{S} \times \frac{N}{S}, S \times S, d_h)$ to reduce the computation

complexity $N^4$ of self-attention to $(\frac{N}{S} \times \frac{N}{S}) \times (S \times S)^2 = N^2 \times S^2$, where $S$ denotes window size. Meanwhile, the computation complexity of the global interaction layer ($N^2 \times M$) is negligible, as the number of global tokens $M$ is much smaller than the window size $S^2$ in our method.

### 3.4 Iterative Learning

To alleviate the noisy mining process, we further propose an iterative learning strategy to enable global structure information and entity embeddings mutually reinforce each other. Specifically, we incorporate global structure knowledge into entity representations through a gating mechanism:

$$
\begin{aligned}
g &= \text{sigmoid}(W_g[H_{k/v}^{(t)}; \widehat{H}_{k/v}^{(t+1)}] + b_g) \\
H_{k/v}^{(t+1)} &= H_{k/v}^{(t)} + g \cdot \widehat{H}_{k/v}^{(t+1)}
\end{aligned}
\tag{10}
$$

Finally, these new key and value features are fed back to the classifier for the next iteration. After repeating this iterative process $K$ times, we get updated key and value features $H_{k/v}^{(K)}$ to compute final logits $l^{(K)}$. Finally, we calculate Binary Cross Entropy (BCE) loss based on $l^{(K)}$ as follows [2]:

$$
\mathcal{L} = \sum_{i=1}^{N} \sum_{j=1}^{N} \ell(l_{i,j}^{(K)}, y_{i,j})
\tag{11}
$$

where $y_{i,j} \in [0, 1]$ is binary ground truth of the entity pair $(e_i, e_j)$, $\ell(.,.)$ is the cross-entropy loss.

## 4 Experiments

In this section, we perform detailed experiments to demonstrate the effectiveness of our proposed method GOSE among different settings. Besides the standard setting of typical language-specific fine-tuning (section 4.2), we further consider more challenging settings to demonstrate the generalizability of GOSE such as cross-lingual zero-shot transfer learning (section 4.3) and few-shot learning (section 4.4). Before discussing the results, we provide the details of the experimental setup below.

### 4.1 Experimental Setup

#### 4.1.1 Datasets

**FUNSD** (Jaume et al., 2019) is a scanned document dataset for form understanding. It has 149 training samples and 50 test samples with various layouts.

---

[2]To make a fair comparison with baselines, we use the same binary classification training strategy.

| Structure Information | Model | FUNSD | XFUND | | | | | | | Avg. |
|---|---|---|---|---|---|---|---|---|---|---|
| | | EN | ZH | JA | ES | FR | IT | DE | PT | |
| Text-only | XLM-RoBERTa | 0.2659 | 0.5105 | 0.5800 | 0.5295 | 0.4965 | 0.5305 | 0.5041 | 0.3982 | 0.4769 |
| | InfoXLM | 0.2920 | 0.5214 | 0.6000 | 0.5516 | 0.4913 | 0.5281 | 0.5262 | 0.4170 | 0.4910 |
| Local | LayoutXLM | 0.5483 | 0.7073 | 0.6963 | 0.6896 | 0.6353 | 0.6415 | 0.6551 | 0.5718 | 0.6432 |
| | XYLayoutLM | - | 0.7445 | 0.7059 | 0.7259 | 0.6521 | 0.6572 | 0.6703 | 0.5898 | - |
| | LiLT | 0.6276 | 0.7297 | 0.7037 | 0.7195 | 0.6965 | 0.7043 | 0.6558 | 0.5874 | 0.6781 |
| Global | $GOSE_{LayoutXLM}$ | 0.5926 | 0.8631 | **0.8258** | 0.8375 | 0.7729 | 0.8035 | 0.7780 | 0.7026 | 0.7720 (+12.88%) |
| | $GOSE_{LiLT}$ | **0.7697** | **0.8752** | 0.8096 | **0.8595** | **0.8646** | **0.8415** | **0.8023** | **0.7384** | **0.8201** (+14.20%) |

Table 1: Language-specific fine-tuning F1 accuracy on FUNSD and XFUND (fine-tuning on X, testing on X)."Text-only" denotes pre-trained textual models without structure information, "Local" denotes pre-trained VrDU models with local features, and "Global" denotes using global structure information.

| Structure Information | Model | FUNSD | XFUND | | | | | | | Avg. |
|---|---|---|---|---|---|---|---|---|---|---|
| | | EN | ZH | JA | ES | FR | IT | DE | PT | |
| Text-only | XLM-RoBERTa | 0.2659 | 0.1601 | 0.2611 | 0.2440 | 0.2240 | 0.2374 | 0.2288 | 0.1996 | 0.2276 |
| | InfoXLM | 0.2920 | 0.2405 | 0.2851 | 0.2481 | 0.2454 | 0.2193 | 0.2027 | 0.2049 | 0.2423 |
| Local | LayoutXLM | 0.5483 | 0.4494 | 0.4408 | 0.4708 | 0.4416 | 0.4090 | 0.3820 | 0.3685 | 0.4388 |
| | LiLT | 0.6276 | 0.4764 | 0.5081 | 0.4968 | 0.5209 | 0.4697 | 0.4169 | 0.4272 | 0.4930 |
| Global | $GOSE_{LayoutXLM}$ | 0.5926 | 0.5696 | 0.5556 | 0.5124 | 0.5295 | 0.4168 | 0.4325 | 0.4363 | 0.5056 (+6.68%) |
| | $GOSE_{LiLT}$ | **0.7697** | **0.6930** | **0.6805** | **0.7072** | **0.7145** | **0.6355** | **0.5997** | **0.5830** | **0.6729** (+17.99%) |

Table 2: Cross-lingual zero-shot transfer F1 accuracy on FUNSD and XFUND (fine-tuning on FUNSD, testing on X).

**XFUND** (Xu et al., 2021) is a multilingual form understanding benchmark. It includes 7 languages with 1,393 fully annotated forms. Each language includes 199 forms. where the training set includes 149 forms, and the test set includes 50 forms.

### 4.1.2 Baselines

We use the following baselines: (1) text-only pre-trained models without structure information: XLM-RoBERT (Conneau et al., 2020), InfoXLM (Chi et al., 2021); (2) layout-aware pre-trained VrDU models with local structure information: LayoutXLM (Xu et al., 2021), XYLayoutLM (Gu et al., 2022), LiLT (Wang et al., 2022a). All of the experimental results of these baselines are from their original papers directly, except for the results on the few-shot learning task [3].

### 4.1.3 Experiment Implementation

We use the bi-affine classifier and binary classification training loss over all datasets and settings following (Wang et al., 2022a) for a fair comparison. The entity representation is the first token vector in each entity. For the few-shot learning task, we randomly sample training samples over each shot five times with different random seeds,

and report the average performance under five sampling times for a fair comparison. More details of the training hyper-parameters can be found in Appendix A.

### 4.2 Language-specific Fine-tuning

We compare the performance of GOSE applied to the language-specific fine-tuning task. The experimental results are shown in Table 1. First, all VRE methods with structure information outperform the text-only models XLM-RoBERT and InfoXLM, which indicates structure information plays an important role in the VrDU. Second, while pre-trained VrDU models have achieved significant improvement over text-only models, our method still outperforms them by a large margin. This phenomenon denotes that incorporating global structure information is generally helpful for VRE. Compared to the SOTA method LiLT (Wang et al., 2022a), GOSE achieves significant improvements on all language datasets and has an increase of 14.20% F1 accuracy on the average performance. Third, we further observe that our GOSE is model-agnostic, which can consistently improves diverse pre-trained models' relation extraction performance on all datasets. For example, GOSE has an improvement of 12.88% F1 accuracy on average performance compared to LayoutXLM (Xu et al., 2021).

---

[3]We re-implement the results using the official code.

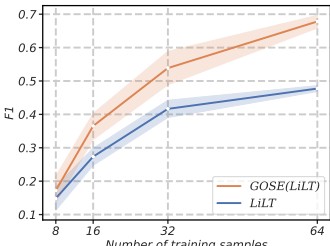

Figure 3: Average performances of few-shot setting on FUNSD dataset. We report the results of random sampling five times over each shot.

| Method | Components | | | F1 Accuracy↑ | |
|---|---|---|---|---|---|
| | GIL | Spatial-Prefix | GSKM | EN | ZH |
| GOSE | ✓ | ✓ | ✓ | **0.7697** | **0.8752** |
| 1  w/o GIL | ✗ | ✓ | ✓ | 0.7384 | 0.8643 |
| 2  w/o Spatial-Prefix | ✓ | ✗ | ✓ | 0.7029 | 0.8161 |
| 3  w/o GSKM | ✗ | ✗ | ✗ | 0.6902 | 0.8037 |

Table 3: Ablation study of our model using LiLT as the backbone on the FUNSD and XUND (ZH). The symbol EN denotes FUNSD and ZH means chinese language.

structure knowledge can improve the generalization of the model.

## 4.3 Cross-lingual Transfer Learning

We evaluate GOSE on the cross-lingual zero-shot transfer learning task. In this setting, the model is only fine-tuned on the FUNSD dataset (in English) and evaluated on each specific language dataset. We present the evaluation results in Table 2. It can be observed that GOSE significantly outperforms its competitors and consistently improve diverse backbone encoders' relation extraction performance on all datasets. This verifies that GOSE can capture the common global structure information invariance among different languages and transfer it to other languages for VRE. We observe that the performance improvement of GOSE(LayoutXLM) is not as significant as GOSE(LiLT). This may be attributed to that the architecture of LiLT decoupling the text and layout information makes it easier to learn language-independent structural information, which is consistent with the observation of previous works (Wang et al., 2022a). We further evaluate GOSE on the Multilingual learning task, the results are shown in Appendix B.

## 4.4 Few-shot Learning

Previous experiments illustrate that our method achieves improvements using full training samples. We further explore whether GOSE could mine global structure information in the low-resource setting. Thus, we compare with the previous SOTA model LiLT on few-shot settings. The experimental results in Figure 3 indicate that the average performance of GOSE still outperforms the SOTA model LiLT. Notably, our GOSE achieves a comparable average performance (67.82%) on FUNSD dataset using only 64 training samples than LiLT does (62.76%) using full training samples, which further proves that our proposed method can more efficiently leverage training samples. This success may be attributed to the incorporation of global

## 4.5 Ablation Study

**Effectiveness of individual components.** We further investigate the effectiveness of different modules in our method. we compare our model with the following variants in Table 3.

(1) *w/o GIL*. In this variant, we remove the global interaction layer from GSKM. This change means that the GSKM module only performs local self-attention. The results shown in Table 3 suggest that our GIL can encourage the GSKM module to better exploit dependencies of entity pairs.

(2) *w/o Spatial-Prefix*. In this variant, we remove the spatial prefix-guided interaction. This change causes a significant performance decay. This suggests the injection of additional spatial layout information can guide the attention mechanism attending to entity pairs with similar spatial structure and thus help the model learn global structural knowledge.

(3) *w/o GSKM*. In this variant, we remove the GSKM module from GOSE. This change means the model only obtains context-aware relation features through a mean-pooling operation. The results shown in Table 3 indicate that although the mean-pooling operation can achieve a performance improvement, the GSKM module can mine more useful global structure information.

**Ablation of Iteration Rounds.** The highlight of our GOSE is mining global structure knowledge and refining entity embeddings iteratively. We argue that these two parts can mutually reinforce each other: the integration of global structure knowledge can help refine entity embeddings. On the contrary, better entity embeddings can help mine more accurate global structure knowledge. Thus, we evaluate the influence of the iteration rounds. The results are shown in 5(a), which indicates GOSE usually achieves the best results within small number of

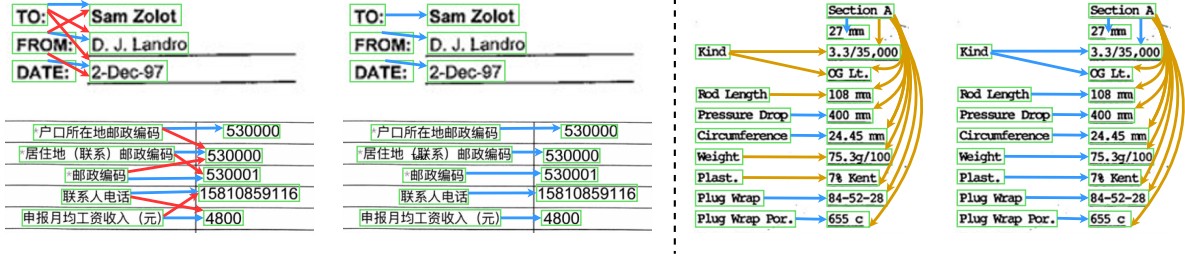

(a) Conflicted Links          (b) Missing Links

Figure 4: The visualization of examples in FUNSD and XFUND(ZH). The left of each example is LiLT and the right is GOSE(LiLT). The arrows in blue, red and orange denote true positive, false positive and false negative (missed) relations respectively. Best viewed by zooming up.

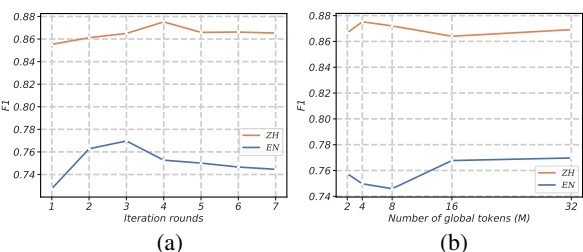

Figure 5: F1 performances of our model using LiLT as the backbone. The symbol EN denotes FUNSD dataset and ZH means XFUND(in chinese). **(a)** Ablation of iteration rounds. **(b)** Ablation of global tokens.

iteration rounds. In addition, we investigate the performance of different information under multiple iterations in Appendix C.

**Ablation of Global Tokens.** We further investigate the effect of the number of global tokens in the GIL on our model. The results are shown in Figure 5 (b), which denotes GOSE can achieve optimal results within a small number of global tokens, while keeping efficiency.

### 4.6 Further Analysis

**Case Study.** To better illustrate the effectiveness of global structure knowledge, we conduct the specific case analysis on the VRE task as shown in Figure 4. Through the visualization of examples, we can notice: (1) as shown in Figure 4(a), GOSE can greatly mitigate the prediction of conflicted links which reveals that our method can capture global structure knowledge to detect conflicted interactions between entity pairs. (2) as shown in Figure 4(b), GOSE can learn long-range relations by analogy with the linking pattern of entity pairs, while keeping a good recall. Notably, it is also difficult for GOSE to predict long-range relations where is not sufficient global structure knowledge. For example, GOSE does not predict well relations

of the entity "*Section A*", due to there are few top-to-bottom and one-to-many linking patterns.

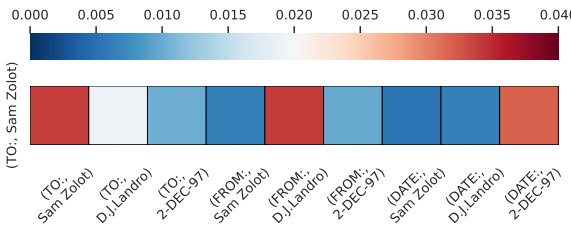

Figure 6: The attention weight visualization between the entity pair (TO:, Sam Zolot) and other spatial prefix of entity pairs.

**Visualization of Attention over Spatial Information.** To illustrate the effect of our proposed spatial prefix-guided local self-attention. We calculate the attention scores over the spatial information for the document in Figure 5(a), i.e., the spatial-prefix guided attention weights using Equation (6). As shown in Figure 6, the entity pair (TO:, Sam Zolot) pays more attention towards the entity pair (FROM:, D.J.Landro) and (DATE:, 2-DEC-97). This phenomenon indicates that the injection of additional spatial layout information of entity pairs can guide the attention mechanism attending to entity pairs with similar spatial structure, thereby enhancing the capacity of the model to discern precise dependencies between entity pairs.

## 5 Related Works

### 5.1 Visual Relation Extraction

Visual relation extraction (VRE) aims at identifying relations between semantic entities from visually-rich documents. Early methods based on graph neural networks (Zhang et al., 2021b; Tang et al., 2021; Li et al., 2021b, 2022b, 2023c) learned node features by treating semantic entities as nodes in

a graph. Recently, most studies (Li et al., 2020; Wang et al., 2022a; Gu et al., 2022; Huang et al., 2022b), used the self-supervised pre-training and fine-tuning techniques to boost the performance on document understanding. Although they have achieved significant improvement on VIE tasks, especially on the semantic entity recognition (SER) task (Peng et al., 2022). VRE remains largely underexplored and is also a challenging task. In this paper, we focus on mining global structure knowledge to guide relation extraction. To the best of our knowledge, our work is the first attempt to exploit dependencies of entity pairs for this task.

## 5.2 Efficient Transformers

Efficient transformers (Dong et al., 2019; Li et al., 2019; Tay et al., 2020; Beltagy et al., 2020; Ryoo et al., 2021; Zhang et al., 2021a; Li et al., 2022c) are a class of methods designed to address the quadratic time and memory complexity problems of vanilla self-attention. More similar to us are methods that leverage down-sampling to reduce the resolution of the sequence, such as window-based vision transformers (Liu et al., 2021; Yang et al., 2021; Huang et al., 2022a; Wang et al., 2022b). Different from existing methods, we propose the spatial prefix-guided attention mechanism, which leverages spatial layout properties of VrDs to guide GOSE to mine global structure knowledge.

## 6 Discussion

Recently, The Multimodal Large Language Model (MLLM) (Li et al., 2022a; Yin et al., 2023) has emerged as a pioneering approach. MLLMs leverage powerful Large Language Models (LLMs) as a brain to perform multimodal tasks. The surprising emergent capabilities of MLLM, such as following zero-shot demonstrative instructions (Li et al., 2023b) and OCR-free math reasoning (Zhu et al., 2023; Dai et al., 2023), are rare in traditional methods. Several studies (Xu et al., 2023; Liu et al., 2023) have conducted comprehensive evaluations of publicly available large multimodal models. These investigations reveal that MLLMs still struggle with the VIE task. In this paper, we introduce and empirically validate that global structural knowledge is useful for visually-rich document information extraction. Our insights have the potential to shape the advancement of large model technology in the domain of visually-rich documents.

## 7 Conclusion

In this paper, we present a general global structure knowledge-guided relation extraction method for visually-rich documents, which jointly and iteratively learns entity representations and mines global dependencies of entity pairs. To the best of our knowledge, GOSE is the first work leveraging global structure knowledge to guide visual relation extraction. Concretely, we first use a base module that combines with a pre-trained model to obtain initial relation predictions. Then, we further design a relation feature generation module that generates relation features and a global structure knowledge mining module. These two modules perform the "generate-capture-incorporate" process multiple times to mine and integrate accurate global structure knowledge. Extensive experimental results on three different settings (e.g., standard fine-tuning, cross-lingual learning, low-resource setting) over eight datasets demonstrate the effectiveness and superiority of our GOSE.

## Acknowledgments

We would like to express gratitude to the anonymous reviewers for their kind comments. This work has been supported in part by the Zhejiang NSF (LR21F020004), Key Research and Development Projects in Zhejiang Province (No. 2023C01030, 2023C01032), NSFC (No. 62272411), National Key Research and Development Program of China (2018AAA0101900), Ant Group and Alibaba-Zhejiang University Joint Research Institute of Frontier Technologies. Our work was also supported by Scientific Research Fund of Zhejiang Provincial Education Department.

## Limitations

The proposed work still contains several limitations to address in future work, as follows:

**Method.** One limitation of our method is that it cannot capture global structure information from informative visual clues. The visual features in visually-rich documents such as font size and color may provide diverse structure information. For example, section titles in resumes and job ads are often in fonts different from the content. We leave this for future work.

**Task.** We only evaluate the visual relation extraction task covering diverse settings. Due to the limited budget and computation resources, we cannot

afford evaluation on more tasks related to visually-rich documents. We will plan to evaluate the proposed approach on more visual information extraction tasks such as semantic entity recognition.

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

## A Hyperparameters

All of our experiments are performed on one NVIDIA 3090 GPU with the PyTorch (Paszke et al., 2019) framework. We use a mini-batch AdamW (Loshchilov and Hutter, 2018) optimizer with a weight decay of 0.1. The model is trained with a batch size of 2. The window size is fixed to 64. In the Language-Specific Fine-tuning experiments for all languages, the learning rate, steps are set to $6.25 \times 10^{-6}, 2 \times 10^4$ accordingly on LiLT encoder, while $2.5 \times 10^{-5}, 4 \times 10^4$ on the Layoutxlm encoder. In the multilingual learning experiments, we use the full language-specific data for training, with total steps $1.6 \times 10^5$. In the cross-lingual zero-shot transfer learning experiments, we directly evaluate the model, which was trained in the previous language-specific experiments, on the XFUND dataset.

## B Multilingual Learning

We evaluate GOSE on the multilingual learning setting. In this setting, the model is fine-tuned with all 8 languages simultaneously and evaluated on each specific language. From the experimental results shown in Table 4, we can find that although this setting further improves the baseline model performance compared to the language-specific fine-tuning, our method GOSE once again outperforms its counterparts by a large margin. We hold that the superior performance of our method GOSE can be attributed to the fact that the previous method still does not learn sufficient global structure information in multilingual learning. This finding also demonstrates that mining global structure information is beneficial for the VRE task.

| | Model | FUNSD | XFUND (Avg.) |
|---|---|---|---|
| Text-only | XLM-RoBERTa | 0.3638 | 0.6727 |
| | InfoXLM | 0.3699 | 0.6495 |
| Local | LayoutXLM | 0.6671 | 0.7988 |
| | LiLT | 0.7407 | 0.8228 |
| Global | GOSE$_{LayoutXLM}$ | 0.7755 | 0.8656 |
| | GOSE$_{LiLT}$ | **0.9003** | **0.8994** |

Table 4: Multilingual fine-tuning F1 accuracy on FUNSD and XFUND (fine-tuning on 8 languages all, testing on X).

## C Effect of Different Information

We show the performance of our GOSE (LiLT) under multiple iterations in Figure 7. We can observe

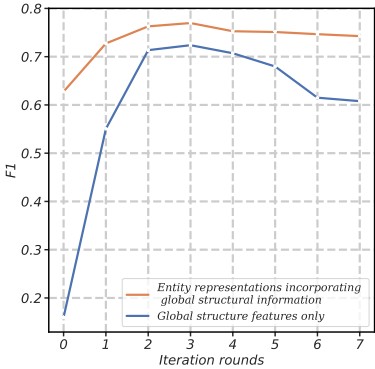

Figure 7: F1 performances of different information of our GOSE (LiLT) under multiple iterations on FUNSD dataset.

that (1) when the iterative learning process begins, both global structural features and entity representations progressively enhance and mutually reinforce each other. (2) In scenarios where the number of iteration rounds stands at zero, i.e., without iterative learning. The performance of global structure information is poor. This may be because the initial entity representations obtained from the pre-trained model are not strong, thus the mined structural information without iterative optimization is noisy. (3) As the number of iteration rounds increases to a certain point, the performance of the model decreases. This phenomenon can be attributed to too many iteration rounds that can cause mined global structural information to become over-smoothing thus affecting the final performance of our model.