# OpenReview forum: "Global Structure Knowledge-Guided Relation Extraction Method for Visually-Rich Document"
_EMNLP/2023/Conference — EMNLP 2023 Findings_

### Official Review · Reviewer_5V4t · 2023-07-21

**Soundness:** 3

**Excitement:**

3: Ambivalent: It has merits (e.g., it reports state-of-the-art results, the idea is nice), but there are key weaknesses (e.g., it describes incremental work), and it can significantly benefit from another round of revision. However, I won't object to accepting it if my co-reviewers champion it.

**Paper Topic And Main Contributions:**

This paper explores the long-neglected global spatial/structure information in the VRE task by tackling two technical challenges, proposes GOSE to incorporate the information, and demonstrates the advantages of GOSE such as data efficiency, cross-lingual transfer, and SOTA performance.

**Questions For The Authors:**

Q1: I notice that the extracted relations are tuples instead of triplets since the head entity itself actually represents a certain kind of relation. Is there any work in VRE that extracts relational triplets from the scanned images of documents? Are there any human-annotated datasets?
Q2: Perhaps the sharp performance drop (without Iteration) in Sec. 4.5 can originate from the significant decline of model parameters?

**Reasons To Accept:**

The experimental results are sufficient to demonstrate both the effectiveness and efficiency of GOSE.
The paper is well-written and clearly presents the motivation and ideas of GOSE.

**Reasons To Reject:**

The paper does not discuss the performance of LLMs that are able to process multi-modal information (scanned image of a document coupled with a prompt) and then provide useful information, which renders it unclear to further estimate the paper’s contribution to the VRE task.
The paper argues that the main contribution of GOSE is that dependencies between entity pairs are accurately captured and modeled, but it seems that more experiments are required to provide the evidence. It remains unclear for me about how the dependencies can work and boost the performance. I think, for example, it will be a bonus if authors visualize the attention weights over the spatial information (e.g., spatial prefix).
The reproducibility of the paper may be a concern, as the authors did not mention whether the code or data will be publicly available for others to replicate or build upon this work.

**Reproducibility:**

3: Could reproduce the results with some difficulty. The settings of parameters are underspecified or subjectively determined; the training/evaluation data are not widely available.

**Reviewer Confidence:**

3: Pretty sure, but there's a chance I missed something. Although I have a good feel for this area in general, I did not carefully check the paper's details, e.g., the math, experimental design, or novelty.

---

> ### Author Rebuttal · Authors · 2023-08-29
>
> ## Response to Reviewer 5V4t
>
> We appreciate the reviewer for the valuable comments. Our response to the reviewer’s questions is as follows.
>
> > **Q1:** The paper does not discuss the performance of LLMs that are able to process multi-modal information (scanned image of a document coupled with a prompt) and then provide useful information, which renders it unclear to further estimate the paper’s contribution to the VRE task.
>
> **A1:** Thanks for raising this important consideration. Recent studies [E, F] have conducted comprehensive evaluations of publicly available large multimodal models. These  investigations reveal that large multimodal models primarily rely on semantic understanding for complex reasoning and exhibit inferior perception of fine-grained structure information without semantic. For example, for the VQA task,  InstructBLIP [G] achieves an impressive top-1 accuracy of 60.20% on the OCR-VQA benchmark. However, for the key information extraction (KIE) task, which consists of semantic entity recognition (SER) and visual relationship extraction (VRE), large multimodal models demonstrate notably poor performance. On the FUNSD benchmark which is used in our paper, InstructBLIP  achieves 1.03% F1 score,  MiniGPT-4 [H]  achieves 1.27% F1 score, and mPLUG-Owl [I] achieves 0.41% F1 score.  Furthermore, our paper introduces and empirically validates that the global structural knowledge is useful for visually-rich document information extraction. We firmly believe that our insights have the potential to guide the development of large model technology within the visually-rich document domain. As you nicely suggested, we will include this discussion in our paper.
>
> [E] On the Hidden Mystery of OCR in Large Multimodal Models. Liu et al. Arxiv 2023.
>
> [F] LVLM-eHub: A Comprehensive Evaluation Benchmark for Large Vision-Language Models. Xu et al. Arxiv 2023.
>
> [G] InstructBLIP: Towards General-purpose Vision-Language Models with Instruction Tuning. Dai et al. Arxiv 2023.
>
> [H] MiniGPT-4: Enhancing Vision-Language Understanding with Advanced Large Language Models. Zhu et al. Arxiv 2023.
>
> [I] mPLUG-Owl: Modularization Empowers Large Language Models with Multimodality. Ye et al. Arxiv 2023.
>
>
>
>
>
> > **Q2:** The paper argues that the main contribution of GOSE is that dependencies between entity pairs are accurately captured and modeled, but it seems that more experiments are required to provide the evidence. It remains unclear for me about how the dependencies can work and boost the performance. I think, for example, it will be a bonus if authors visualize the attention weights over the spatial information (e.g., spatial prefix).
>
> **A2:**  Thanks for your important concern. We calculate the attention scores over the spatial information for the document in Figure 5(a), i.e., the spatial-prefix guided attention weights using Equation (6). We provide the results of attention weights over the spatial prefix in the following table. As shown in the following table,  the entity pair (TO, Sam Zolot)  pays more attention towards the entity pair (FROM:, D.J.Landro) and  (DATE:, 2-DEC-97).  This phenomenon indicates that the injection of additional spatial layout information of entity pairs can guide the attention mechanism attending to entity pairs with similar spatial structure, thereby enhancing the capacity of the model to discern precise dependencies between entity pairs. We will visualize these attention weights in the next version.
>
> i.  The spatial-prefix guided attention weights of the entity pair (TO, Sam Zolot) over other spatial prefix of entity pairs.
>
> |                 | (TO, Sam Zolot) | (TO, D.J.Landro) | (TO, 2-DEC-97) | (FROM:, Sam Zolot) | (FROM:, D.J.Landro) | (FROM:, 2-DEC-97) | (DATE:, Sam Zolot) | (DATE:, D.J.Landro) | (DATE:, 2-DEC-97) |
> | :-------------: | :-------------: | :--------------: | :------------: | ------------------ | ------------------- | ----------------- | ------------------ | ------------------- | ----------------- |
> | (TO, Sam Zolot) |   **0.0189**    |      0.0338      |     0.0318     | 0.0064             | **0.0097**          | 0.0340            | 0.0054             | 0.0064              | **0.0103**        |
>
>
>
>
>
> >**Q3:** The reproducibility of the paper may be a concern, as the authors did not mention whether the code or data will be publicly available for others to replicate or build upon this work.
>
> **A3:** Thanks for your concern. We have provided the data and code as part of the Supplementary Materials.  Additionally, we will include a statement within our paper affirming our commitment to making all data and code publicly accessible. We greatly value the opportunity to enhance the transparency and reproducibility of our research through these actions.
>
>
>
>
>
>
>
> > **Q4:** I notice that the extracted relations are tuples instead of triplets since the head entity itself actually represents a certain kind of relation. Is there any work in VRE that extracts relational triplets from the scanned images of documents? Are there any human-annotated datasets?
>
> **A4:** Thanks for your question. We appreciate your proper understanding of the visually-rich document relation extraction task. This task entails determining the presence of relationships between two semantic entities and subsequently extracting tuples comprising related entity pairs. Given the inherent definition of the VRE task, prevailing methodologies have predominantly directed their efforts towards effectively capturing and modeling the multimodal information embedded in scanned document images to achieve precise tuple extraction [J]. Due to fine-grained annotation requirements and the poor performances of LLMs in the visually-rich document domain [E, F], current high-quality datasets require manual annotation, which encompasses the FUNSD and XFUND datasets used in our paper.  Therefore,  Using machines to automatically annotate visually-rich documents may be a challenging and interesting topic. We appreciate that the reviewer points out a meaningful and interesting direction for future research.  We leave this for future work.
>
> [J] DOCUMENT AI: BENCHMARKS, MODELS AND APPLICATIONS. Cui et al. Arxiv 2021.
>
>
>
>
>
> > **Q5:** Perhaps the sharp performance drop (without Iteration) in Sec. 4.5 can originate from the significant decline of model parameters?
>
> **A5:**  Thanks for your question.  The sharp performance drop (without Iteration) in Sec. 4.5 is mainly due to the fact that we only use the mined global structural information to make predictions without fusing entity representations (using Equation (9)). Since the initial entity representation obtained from the pretrained model is not strong, the mined structural information without iterative optimization is noisy.  The iterative learning process can refine the mined global structural information and enhance the richness of entity representations. This concerted effort leads to a notable enhancement in the performance of model.
>
> To illustrate this point more clearly, We extract a subset of the training dataset as a validation dataset to perform the hyper-parameter tuning. When tuning the iteration rounds, we keep other hyper-parameters constant. We add the experiment of different information of our GOSE (LiLT) under multiple iterations over FUNSD dataset in the following table.
>
> |                         Information                          |                    Rounds 0                     | Rounds 1 | Rounds 2 |  Rounds 3  | Rounds 4 | Rounds 5 | Rounds 6 | Rounds 7 |
> | :----------------------------------------------------------: | :---------------------------------------------: | :------: | :------: | :--------: | :------: | :------: | :------: | :------: |
> |                Global structure features only                |                     0.1599                      |  0.5505  |  0.7135  | **0.7238** |  0.7071  |  0.6797  |  0.6150  |  0.6078  |
> | Entity representations  incorporating global structural information | 0.6276*  |  0.7274  |  0.7628  | **0.7697** |  0.7527  |  0.7511  |  0.7466  |  0.7426  |
>
> As shown in above table, the first line demonstrates the performance of only using mined global structural information, the second line demonstrates the performance of using entity representations incorporating global structural information. We can observe that:
>
> (1) When iterative learning process begins, both global structural features and entity representations progressively enhance and mutually reinforce each other. For example, by executing one iteration round (rounds = 1), the performance of only global structure features improves from 0.1599 (absent iteration) to 0.5505. Similarly, the performance of entity representations advances from 0.6276 (without global structural information, rounds=0) to 0.7274 (incorporating global structural information, rounds=1).
>
> (2) As the number of iteration rounds increases to a certain point, the performance of the model decreases. This phenomenon can be attributed to too many iteration rounds can cause mined global structural information to become over-smoothing thus affecting the final performance of our model.

---

### Official Review · Reviewer_ZM3X · 2023-08-04

**Typos Grammar Style And Presentation Improvements:** "). VRE remains" --> "), VRE remains"…
**Soundness:** 3

**Excitement:**

3: Ambivalent: It has merits (e.g., it reports state-of-the-art results, the idea is nice), but there are key weaknesses (e.g., it describes incremental work), and it can significantly benefit from another round of revision. However, I won't object to accepting it if my co-reviewers champion it.

**Paper Topic And Main Contributions:**

This paper studied Visual Relation Extraction (VRE) which aims to extract relationships between entities from visually-rich documents.
The authors proposed a GlObal Structure knowledge-guided relation Extraction (GOSE) framework which makes use of both local spatial layout and global structure information to extract relations, and use an iterative learning method to learning the most informative features. The experimental results show good performance of the proposed method.

**Questions For The Authors:**

This paper presents very impressive results, with huge improvement over baseline methods. However, from the ablation study, the most useful part is the iterative learning. The performance of the method surprisingly dropped from F1=0.7697 to 0.1599 when removing the iteration steps. This raises concern on over-fitting, as multiple rounds of iterations can result in over-parameterization and overfitting, particularly when considering that the datasets used in the studies are relatively small and there seems no validation datasets for hyperparameter tuning. It would be useful to extract a subset of a training dataset as a validation dataset from to eliminate this concern.

Some minor comments are:
1. As one of the most important hyper-parameter, It is unclear how many iterations were used in the experiments. It might not be unexpected to see that the best performance was achieved within a small number of iterations, as a large number of iterations can diminish representational power of spatial information. Also, how are the global tokens T initialized?

2. I wonder how the method would deal with entities that spread across multiple pages in a document, and how to adjust the distance and other functions for these cases.

3. It is great to see that the proposed GOSE method is plug-and-play. In experiments, GOSE was used with LayoutXLM or LiLT. It would be useful to mention how GOSE can be plugged with other model in the Method section.

**Reasons To Accept:**

Good results, rich details, and reasonable methods

**Reasons To Reject:**

Suspicious improvement through iterative learning steps;
relatively small datasets;
missing some details in experimental settings.

**Reproducibility:**

3: Could reproduce the results with some difficulty. The settings of parameters are underspecified or subjectively determined; the training/evaluation data are not widely available.

**Reviewer Confidence:**

4: Quite sure. I tried to check the important points carefully. It's unlikely, though conceivable, that I missed something that should affect my ratings.

---

> ### Author Rebuttal · Authors · 2023-08-29
>
> ## Response to Reviewer ZM3X
>
> We sincerely thank you for your comprehensive comments and constructive advice. We will explain your concerns point by point.
>
> > **Q1:** Suspicious improvement through iterative learning steps. This paper presents very impressive results, with huge improvement over baseline methods. However, from the ablation study, the most useful part is the iterative learning. The performance of the method surprisingly dropped from F1=0.7697 to 0.1599 when removing the iteration steps. This raises concern on over-fitting, as multiple rounds of iterations can result in over-parameterization and overfitting, particularly when considering that the datasets used in the studies are relatively small and there seems no validation datasets for hyperparameter tuning. It would be useful to extract a subset of a training dataset as a validation dataset from to eliminate this concern.
>
> **A1:** Thanks for your question. The performance of the method surprisingly dropped when removing the iteration steps is mainly due to the fact that we only use the mined global structural information to make predictions without fusing entity representations (using Equation (9)). **Since the initial entity representations obtained from the pretrained model is not strong, the mined structural information without iterative optimization is noisy**.  The iterative learning process can refine the mined global structural information and enhance the richness of entity representations. This concerted effort leads to a notable enhancement in the performance of model.
>
> To illustrate this point more clearly, We extract a subset of the training dataset as a validation dataset to perform the hyper-parameter tuning. When tuning the iteration rounds, we keep other hyper-parameters constant. We add the experiment of different information of our GOSE (LiLT) under multiple iterations over FUNSD dataset in the following table.
>
> |                         Information                          | Rounds 0 | Rounds 1 | Rounds 2 |  Rounds 3  | Rounds 4 | Rounds 5 | Rounds 6 | Rounds 7 |
> | ---------------------------------------------------------- | :---------------------------------------------: | :------: | :------: | :--------: | :------: | :------: | :------: | :------: |
> |                Global structure features only                |                     0.1599                      |  0.5505  |  0.7135  | **0.7238** |  0.7071  |  0.6797  |  0.6150  |  0.6078  |
> | Entity representations  incorporating global structural information | 0.6276*  |  0.7274  |  0.7628  | **0.7697** |  0.7527  |  0.7511  |  0.7466  |  0.7426  |
>
> As shown in above table, the first line demonstrates the performance of only using mined global structural information, the second line demonstrates the performance of using entity representations incorporating global structural information. We can observe that:
>
> (1) When iterative learning process begins, both global structural features and entity representations progressively enhance and mutually reinforce each other. For example, by executing one iteration round (rounds = 1), the performance of only global structure features improves from 0.1599 (absent iteration) to 0.5505. Similarly, the performance of entity representations advances from 0.6276 (without global structural information, rounds=0) to 0.7274 (incorporating global structural information, rounds=1).
>
> (2) As the number of iteration rounds increases to a certain point, the performance of the model decreases. This phenomenon can be attributed to too many iteration rounds can cause mined global structural information to become over-smoothing thus affecting the final performance of our model.
>
> In addition to the above observations, we further extract a subset of a training dataset as a validation dataset to hyperparameter tuning to eliminate the raised concern on over-fitting.  Please refer to our reply to the Q4 of you. We will include these discussions in the next version according to your valuable suggestion.
>
>
>
> > **Q2:** Relatively small datasets.
>
> **A2:** We sincerely appreciate your thoughtful feedback on our work. We completely acknowledge the significance of datasets in establishing the generalizability of our results.  Thus, we choose widely recognized and publicly accessible datasets for the purpose of meaningful comparison.  Although the size of each dataset is relatively small,  we integrated FUNSD and XFUND datasets to explore more challenging settings. For instance, we evaluate our GOSE on the cross-lingual zero-shot transfer learning task (in Section 4.3). In this setting, the model is only fine-tuned on the FUNSD dataset and directly evaluated on the XFUND dataset. At the same time, we merged the FUNSD and XFUND datasets into a relatively large dataset to evaluate the multilingual learning task (in Appendix 2). In this setting, the model is fine-tuned with all 8 languages simultaneously and evaluated on each specific language.  These comprehensive settings demonstrate the robustness and effectiveness of our GOSE across diverse contexts. Furthermore, We notice some works [A, B] published results on large-scale yet private datasets. We will conduct experiments on larger datasets when these large-scale dataset are publicly available.
>
> [A] Chargrid: Towards Understanding 2D Documents. Katti et al. EMNLP 2018
>
> [B] Entity Relation Extraction as Dependency Parsing in Visually Rich Documents. Zhang et al. EMNLP 2021.
>
>
>
> >**Q3:** Missing some details in experimental settings.
>
> **A3:** Thanks for the nice suggestion.  As mentioned in Section 4.1.3, we follow the experimental setup in LiLT  [C] for a fair comparison.  We  will further explain details of experimental settings in our paper.
>
> [C] LiLT: A Simple yet Effective Language-Independent Layout Transformer for Structured Document Understanding. Wang et al. ACL 2022.
>
>
>
> > **Q4:** As one of the most important hyper-parameter, It is unclear how many iterations were used in the experiments. It might not be unexpected to see that the best performance was achieved within a small number of iterations, as a large number of iterations can diminish representational power of spatial information. Also, how are the global tokens T initialized?
>
> **A4:** Thanks for your question. We extract a subset of the training dataset as a validation dataset to perform the hyper-parameter tuning. When tuning the iteration rounds, we keep other hyper-parameters constant. The optimal number of iteration rounds is set to 3 on FUNSD and 4 on XFUND respectively. We provide detailed results under different iteration rounds in the following tables. As you correctly understand, as the number of iteration rounds increases to a certain point, the performance of the model decreases.  The global tokens T are initialized zero. We will declare these more clearly in our paper.
>
> i. Results ( F1 accuracy) of GOSE under different iteration rounds over FUNSD dataset.
>
>
>
> | Iteration Rounds |   FUNSD    |
> | :--------------: | :--------: |
> |        1         |   0.7274   |
> |        2         |   0.7628   |
> |        3         | **0.7697** |
> |        4         |   0.7527   |
> |        5         |   0.7511   |
> |        6         |   0.7466   |
> |        7         |   0.7426   |
>
> ii. Results ( F1 accuracy) of GOSE under different iteration rounds over XFUND dataset.
>
> | Iteration Rounds |     ZH     |     JA     |     ES     |     FR     |     IT     |     DE     |     PT     |    Avg.    |
> | :--------------: | :--------: | :--------: | :--------: | :--------: | :--------: | :--------: | :--------: | :--------: |
> |        1         |   0.8553   |   0.7998   |   0.8345   |   0.8116   |   0.8235   |   0.7747   |   0.7107   |   0.8016   |
> |        2         |   0.8612   |   0.8014   |   0.8460   |   0.8371   |   0.8311   |   0.7816   |   0.7156   |   0.8103   |
> |        3         |   0.8650   |   0.8045   |   0.8499   |   0.8408   |   0.8345   |   0.7930   |   0.7224   |   0.8151   |
> |        4         | **0.8752** | **0.8096** | **0.8595** | **0.8646** | **0.8415** | **0.8023** | **0.7384** | **0.8273** |
> |        5         |   0.8659   |   0.8081   |   0.8589   |   0.8554   |   0.8379   |   0.8017   |   0.7378   |   0.8236   |
> |        6         |   0.8662   |   0.8056   |   0.8504   |   0.8488   |   0.8319   |   0.7929   |   0.7296   |   0.8189   |
> |        7         |   0.8653   |   0.8024   |   0.8469   |   0.8425   |   0.8281   |   0.7874   |   0.7236   |   0.8137   |
>
>
>
> > **Q5:** I wonder how the method would deal with entities that spread across multiple pages in a document, and how to adjust the distance and other functions for these cases.
>
> **A5:**  Thanks for raising a concern about the multi-page document scenario. In response to your concern, we propose a feasible solution to tackle the multi-page VRE challenge. Our strategy involves treating this complex task as a sequence generation problem. First, we can use our GOSE to capture relevant global structure information from each page conditioned to the query entity separately. Then relevant global structure information from each page is concatenated to provide a decoder with a holistic structure representation of the multi-page document at the time of generating target entities.  Recently,  we notice that some works have begun to propose multi-page DocVQA benchmarks [D]. In the future, if there is a publicly available multi-page VRE benchmark, we will attempt to employ our GOSE for this task.
>
> [D] Hierarchical multimodal transformers for Multi-Page DocVQA. Tito et el.  Pattern Recognition 2023.
>
>
>
> > **Q6:** It is great to see that the proposed GOSE method is plug-and-play. In experiments, GOSE was used with LayoutXLM or LiLT. It would be useful to mention how GOSE can be plugged with other model in the Method section.
>
> **A6:**  Thanks for the suggestion.  The reason for plug-and-play is that GOSE only needs to receive entity representations as inputs. The entity representations can be derived from any pre-trained model. After receiving entity representations, our GOSE automatically mines the global structural information to enhance the entity representation. As illustrated in Section 3.1,  We can use other models as a visually-rich document encoder to encode the  scanned image of a document to obtain initial semantic entity representations in the same way that LiLT and LayoutXLM are used. We will adjust the presentation to make it easier to understand.

---

### Official Review · Reviewer_5vV3 · 2023-08-05

**Typos Grammar Style And Presentation Improvements:** Please see the "Reasons to Reject" ab…
**Soundness:** 3

**Excitement:**

3: Ambivalent: It has merits (e.g., it reports state-of-the-art results, the idea is nice), but there are key weaknesses (e.g., it describes incremental work), and it can significantly benefit from another round of revision. However, I won't object to accepting it if my co-reviewers champion it.

**Missing References:**

N/A

**Paper Topic And Main Contributions:**

This paper studies visual relation extraction and introduces a novel method to learn the global structure of the document to guide the extraction.  This paper tackles two challenges to effectively mine and incorporate global structure knowledge, including the vast mining space for capturing dependencies between entity pairs and the noisy mining process due to the lack of direct supervision. Experimental results on cross-lingual learning and few-shot learning show the effectiveness of the proposed method.

**Questions For The Authors:**

Please see the "Reasons to Reject" above.

**Reasons To Accept:**

1. The motivation of this paper is clear
2. This paper is well-written and can be easily followed.
3. The proposed local- and global- learning procedure seems novel.
4. Extensive experiments with promising results.


**Reasons To Reject:**

1. Lack of explanations of how the local- and glocal-structure information can be learned, and visualization is suggested to convenience audiences.

2. There are many typos scattered, and careful proofreading is suggested. e.g.
Line 402:   "we random sample training samples"   "random"-->"randomly"
Line 413:    "which indicate structure information plays"   "indicate"-->" indicates"
Line 437:    "It can be observed that GOSE ..... and consistently improve"   "improve"-->"improves"



**Reproducibility:**

4: Could mostly reproduce the results, but there may be some variation because of sample variance or minor variations in their interpretation of the protocol or method.

**Reviewer Confidence:**

3: Pretty sure, but there's a chance I missed something. Although I have a good feel for this area in general, I did not carefully check the paper's details, e.g., the math, experimental design, or novelty.

---

> ### Author Rebuttal · Authors · 2023-08-29
>
> ## Response to Reviewer 5vV3
>
> We sincerely thank you for the valuable comments. We are encouraged to see that our work is recognized as novel and clear. We will explain your concerns point by point.
>
> > **Q1:** Lack of explanations of how the local- and global-structure information can be learned, and visualization is suggested to convenience audiences.
>
> **A1:** In our paper, the term "local structure information" refers to entity representations, which are derived from a pre-trained model serving as the base module. On the other hand, "global structure information" entails the exploration of associations between entity pairs. Notably, the interplay between local and global structure information is mutually reinforcing throughout the iterative learning process. We add an attention weights analysis to better illustrate how our GOSE accurately captures dependencies between entity pairs. We calculate the attention scores over the spatial information for the document in Figure 5(a), i.e., the spatial-prefix guided attention weights using Equation (6). We provide the results of attention weights over the spatial prefix in the following table. As shown in the following table,  the entity pair (TO, Sam Zolot)  pays more attention towards the entity pair (FROM:, D.J.Landro) and  (DATE:, 2-DEC-97).  This phenomenon indicates that the injection of additional spatial layout information of entity pairs can guide the attention mechanism attending to entity pairs with similar spatial structure, thereby enhancing the capacity of the model to discern precise dependencies between entity pairs.  As you nicely suggested, we will include the visualization of attention weights in the next version.
>
> i.  The spatial-prefix guided attention weights of the entity pair (TO, Sam Zolot) over spatial prefix of other entity pairs.
>
> |                 | (TO, Sam Zolot) | (TO, D.J.Landro) | (TO, 2-DEC-97) | (FROM:, Sam Zolot) | (FROM:, D.J.Landro) | (FROM:, 2-DEC-97) | (DATE:, Sam Zolot) | (DATE:, D.J.Landro) | (DATE:, 2-DEC-97) |
> | :-------------: | :-------------: | :--------------: | :------------: | ------------------ | ------------------- | ----------------- | ------------------ | ------------------- | ----------------- |
> | (TO, Sam Zolot) |   **0.0189**    |      0.0338      |     0.0318     | 0.0064             | **0.0097**          | 0.0340            | 0.0054             | 0.0064              | **0.0103**        |
>
> > **Q2:** There are many typos scattered, and careful proofreading is suggested. e.g. Line 402: "we random sample training samples" "random"-->"randomly" Line 413: "which indicate structure information plays" "indicate"-->" indicates" Line 437: "It can be observed that GOSE ..... and consistently improve" "improve"-->"improves"
>
> **A2:** Thank you for your review and guidance. We will proceed with careful proofreading to ensure the submitted paper is grammatically and spelling-wise impeccable.

---

### Meta-Review · Area_Chair_CLmG · 2023-09-15

**Recommendation:** 3

**Metareview:**

The paper targets a practical setting of IE on visually riched documents. Instead of pure text, the inputs can be pdf or some other visual based file. The paper is well motivated and organized, and the experiments show the effectiveness.After serious discussion and consideration, the reviewers' main concerns are about the huge improvements by iterative component, small datasets, and comparison with LLMs. The authors actively provide more evidence which we think has mostly solved the concerns. Nevertheless, some reviewers are not excited about the idea, suggesting the effectiveness yet not much insights.

Overall, we appreciate both the efforts of reviewers and authors. This work is ready to publish, and the only concern is the novelty. We hope the authors can continue to improve the paper according to the comments and your response.

---

### Decision · Program_Chairs · 2023-10-07

**Decision:**

Accept-Findings

**Comment:**

The paper targets a practical setting of IE on visually riched documents. Instead of pure text, the inputs can be pdf or some other visual based file. The paper is well motivated and organized, and the experiments show the effectiveness.After serious discussion and consideration, the reviewers' main concerns are about the huge improvements by iterative component, small datasets, and comparison with LLMs. The authors actively provide more evidence which we think has mostly solved the concerns. Nevertheless, some reviewers are not excited about the idea, suggesting the effectiveness yet not much insights.

Overall, we appreciate both the efforts of reviewers and authors. This work is ready to publish, and the only concern is the novelty. We hope the authors can continue to improve the paper according to the comments and your response.